# Data Sources and Models for Integrated Mobility and Transport Solutions

**DOI:** 10.3390/s24020441

**Published:** 2024-01-11

**Authors:** Pierfrancesco Bellini, Stefano Bilotta, Enrico Collini, Marco Fanfani, Paolo Nesi

**Affiliations:** DISIT Lab, University of Florence, 50139 Florence, Italy; pierfrancesco.bellini@unifi.it (P.B.); stefano.bilotta@unifi.it (S.B.); enrico.collini@unifi.it (E.C.); marco.fanfani@unifi.it (M.F.)

**Keywords:** mobility and transport, data models, data space, big data, data models

## Abstract

The number of data sources and models in the mobility and transport domain strongly proliferated in the last decade. Most formats have been created to enable specific and innovative applications. On the other hand, the available data models present a certain degree of complexity in terms of their integration and management due to partial overlaps, and in most cases, they could be exploited alternatively to implement the same smart and latest innovative solutions. This paper offers an overview of data models, standards and their relationships. A second contribution highlights any possible exploitation of data models for implementing operational processes for city transportation management and for the feeding of simulation and optimization processes that produce other data results in other data models. The final goal in most cases is the monitoring and control of city transport conditions, as well as the tactic and strategic planning of city infrastructure. This work was developed in the context of the CN MOST, a national center of sustainable mobility in Italy, and it is based on exploiting the Snap4City platform.

## 1. Introduction

Globally, more than four billion people live in urban areas, and by 2050, projections indicate that more than two-thirds of the world’s population will live in highly dense cities [1]. Due to these trends, governments and local authorities tend to allocate a greater share of resources and funding toward urban regions. However, despite the manifold opportunities that urban areas offer for income generation, the increasing urbanization has led to several challenges, in particular to mobility and transportation systems in terms of traffic management and urban plans [2], pollutant emissions [3,4], energy distribution, safety/security and thus sustainability.

In the realm of smart cities, Internet of Things (IoT) devices/entities are playing a pivotal role in the collection of real-time data concerning vehicular traffic, and they are contextualized together with the open data released by government, public and/or private entities [5]. Such data offer valuable opportunities for harnessing data-driven methodologies, exploiting big data technologies. Data access gives the opportunity to set up solutions to monitor/analyze urban conditions (operative activities) and enable the possible computation of predictions and in-depth analyses of specific scenarios for tactic and strategic plans. Such solutions exploit and help to improve at least (i) intelligent transport systems (ITS) [6]—i.e., tools able to monitor, control and optimize traffic management, public transport planning, the detection of critical events, etc.; (ii) AVM (automated vehicle monitoring) for public transportation management; and (iii) mobility as a service (MaaS) platforms [7] to facilitate the integration and thus the exploitation of any public transport offer. In most cases, static/quasistatic data, such as the road graph/city structure, public and private transportation, parking, energy recharge locations, etc., are the offering structures where all the other data and flows can be connected. At the same time, dynamic data on traffic status [8] (such as congestion), car accidents, weather conditions, etc., are taken into account to compute optimal routing solutions according to the user’s need, specific assessment and also to plan how to recover from critical events, to plan strategies to improve viability in city areas, etc. [9].

City infrastructures for mobility and transport data management shall offer capabilities to ingest and accommodate a myriad of heterogeneous data sources and formats/models. To this end, in order to guarantee any efficient big data ingestion, handling and exploitation for mobility and transport, the development of specialized software infrastructures is required [10]. Therefore, data aggregation is one of the most complex technical challenges regarding data modeling and their efficient recovery for real-time exploitation for final users. In smart mobility scenarios, this challenge is self-evident in the need to seamlessly integrate diverse data types/models, ranging from structured data suitable for traditional relational databases to unstructured or semistructured data aligning with noSQL databases [11]. Additionally, the necessity to semantically represent relationships among various data entities adds another complexity layer [12] since the goal is to ensure that stored data can be efficiently queried and analyzed, especially to enable any providing business intelligence tools to help decision makers perform smart mobility analysis, management and planning, including drill down and up in space, relationships and time and to provide data to final users on the move. These needs demand the creation of flexible, reliable and efficient data-integrated representations that can accommodate the complexity of any kind of mobility data, coming from multiple sources and formats.

The main contributions of the paper are three-fold to provide:An updated and comprehensive overview (although not exhaustive) of the research and industrial literature about data modeling and types for smart mobility and transport.The main relationships among different data model concepts to highlight what kind of information (data models) can be obtained by processing them in the whole value chain of data on mobility and transport scenarios.Insights that can be derived by the data models and the business and derived in the context of integrated smart mobility and transport systems addressing multiple data models, spaces and types.

In this paper, we present a wide discussion on data related to the mobility domain together with the required relevant data-modeling solutions so as to exploit such information to realize smart mobility frameworks, analytics and decision support systems to help decision makers and mobility-domain stakeholders face both present and future challenges. In Section 2, an overview of the mobility and transport data is provided, while in Section 3, an in-depth review of the standards and formats/models adopted for the domain is presented. The mobility and transport data models are discussed in Section 4, including a high-level description of the components that can be used for a mobility platform to ingest, manage and exploit mobility data for contextual analysis, reconstructions, key-performance-indicators estimation, making predictions, anomaly detection and what-if analysis. Finally, in Section 5, conclusions are drawn.

This work was carried out in the context of the Italian National Center on Sustainable Mobility (MOST) [13], which refers to the Snap4City platform as a collector of mobility and transport data and a framework for a large range of applications.

## 2. Mobility Transport Data Overview

In the context of today’s mobility and transport solutions, three main stakeholders can be identified: Public Administrations (PAs), Mobility Operators (MOs) and final users (FUs). PAs have a fundamental responsibility for the handling, monitoring, regulation, planning and management of the city transport systems. PAs should carry out activities to monitor the mobility infrastructure, composed of the road network, including streets, cycling paths, pedestrian pathways, parking facilities, gates, etc., to assess the infrastructure status and organize possible maintenance operations, as well as to plan any development to respond to novel/changed needs and operative needs. Additionally, PAs have to monitor the MOs to assess if their transport services are adequate with respect to mobility demand and if they are respectful toward specific regulations at least about security, sustainability, accessibility and general contract. The MOs are public and/or private companies that should offer efficient and affordable transport services. They can be involved in several transport modalities; for example, bus, taxi, car renting, car/bike and scooter sharing/pooling, ride hailing (e.g., Uber and Lyft), etc. MOs’ main responsibilities include vehicle management, route planning and/or reservation scheduling, accounting and billing in the agreement with PAs and other stakeholders. FUs are city users, persons using the transport service (they are the demand of mobility): they can be classified as citizens, commuters, tourists, students, city operators or more generally any individual that needs to move from one place to another. Their feedback on the quality of a service is a valuable piece of information to signal problems and inefficiencies and to identify the priority to be addressed by the PAs and the MOs to improve the transportation services offered to them.

It is undoubtful that to enable the required synergies and satisfy the reciprocal needs of the different stakeholders as identified above, collecting and managing accurate information in a timely manner is a fundamental task. For example, FUs must be aware of transport offers including scheduling, prices, delays or service interruptions to properly plan their trips. MOs are interested in being informed about changes in the road network or any enactment of new regulations to quickly modify/plan routes and rides or to provide updated routing information to drivers, also taking into account real-time traffic conditions. PAs must be aware (controlling, monitoring and receiving information) of the real-time status (position, occupancy, velocity, availability, etc.) of vehicles to verify the compliance of the MOs’ offer with respect to their signed contracts and, at the same time, must have a full overview of the status of the road network in terms of traffic congestion (traffic density in the whole road network), level of pollutant emissions, parking status, malfunctioning of traffic lights, car accidents, etc., to respond to unexpected events and to consciously plan/perform changes within the city road structure to improve services. Data are therefore fundamental to make data-driven decisions.

In Figure 1, a schematic representation of the most relevant data-kind groups involved in the mobility and transport domain is presented. They are represented with green boxes and classified into three main groups. As reported in the figure, such data can be exploited to obtain some higher-level information, as represented in the three blocks at the bottom of Figure 1. Such information includes transportation offers, the infrastructure and the real-time status of the environment, together with mobility demands. In the following, a comprehensive description of each data-kind group is presented, showing how such data contributes to obtaining some higher-level information.

The Organization Data group mainly contributes to defining the city infrastructure representation. This group includes statistical and Geographic Information System (GIS) data that can provide information on main mobility attractors as positions of points of interest (POIs) (industrial areas, shopping centers, schools, entertainment areas, etc.), house and civic number distribution, vehicle typology, population, road network, positions of traffic signs and lights, building plant shapes (and with some limitation, even the height of buildings), raster images depicting cadastral or satellite maps and descriptions of areas limited to the traffic. Asynchronous information like events produced by traffic officers to communicate car accidents, street damage, unauthorized processions, etc., or temporary regulations issued by PAs to change road directions or ban an area from all kinds of service or only for some vehicles has to be considered. This possible information can alter both the infrastructure and the real-time status of the city mobility environment.

Most of the above-mentioned static data could be classified as open data, which can be defined as non-privacy-restricted and nonconfidential data produced and freely released by public or private organizations [14]. A large part of the open data sets is distributed by public government authorities—referred to in this case as Open Government Data (OGD). On the other hand, the above-mentioned information is strictly needed to formalize any mobility offer; for example, including bus and tram routes with their schedules, taxis, sharing and ride-hailing services. MOs’ data are therefore included in this group. This second part of the Organization Data is typically produced by private companies that are interested in providing a third party with only the minimal data to enable their business, maybe in competition with other operators. On the contrary, PAs are interested in making transport services as open as possible and thus having all the data for their daily operations and for plans.

The Sensors Data group includes information coming from the IoT network of sensors that can produce a continuous flow of real-time information. Traffic sensors (spires, virtual spires, BT sniffer, laser, etc.) are at the basis of any assessment of traffic conditions [15], such as counting the number of vehicles traveling on the roads (and distinguishing among the different kinds of vehicle), estimating traffic flow, identifying critical conditions and in some cases recognizing car license plates and thus verifying authorization, insurance, computing origin–destination matrices, etc. Other kinds of data are those related to parking sensors, which may communicate slot occupancy [16], and in some cases also may perform a match with the specific car park slot authorization. In most cases, automatic gates with cameras are used to count entities crossing specific areas, verify plates and authorizations and issue sanctions/taxations for entrance in restricted traffic zones and/or entering the city (for example for tourist buses). Other kinds of sensors can be used to assess weather and environmental/pollutant conditions; they all may contribute to the assessment of real-time status concerning the mobility environment since traffic may produce CO_2_, NO_2_ and other specific pollutants [17]. Additional sensors may be positioned to assess the specific flows of bikes, pedestrians, public and touristic buses, etc.

It is noteworthy that, in an urban context, the infrastructure representation together with its real-time status, if augmented with 3D representations, defines the so-called City Information Modeling (CIM) [18], an extension applied to the city of the concept of Building Information Modeling (BIM), the fundamental building block of smart city Digital Twins [19,20].

The Vehicle and People data, the third group, are typically produced from people and vehicles considering them as devices. They include mobile devices; smart cards used to obtain access to public transport; on-board units (OBUs) installed on cars, buses and other vehicles; mobile cell tower connections; the geolocalization of smartphones and social media posts; as well as any usage data from MaaS services as mobile apps. These data help us understand the mobility patterns of FUs and can be used to compute vehicle/people origin–destination matrices (ODMs). ODMs can be regarded as a formalization of the mobility demand, which is key information to plan and improve mobility services, i.e., the offer. Recent technologies, though not yet widely deployed (represented as the green dashed block in Figure 1), such as Cooperative, Connected and Automated Mobility (CCAM) [21] and Connected and Automated Vehicles (CAV), could hopefully contribute in the near future to better describe any user mobility patterns. Despite the relevance of mobility demand knowledge, it is typically quite difficult to acquire data to produce ODMs modeling the full spectrum of demand due to privacy concerns as regulated by the GDPR [22]. Therefore, alternative solutions to compute approximated ODMs may exploit different kinds of data, for example, IoT; vehicle density; cellular network data; census data; data coming from public transportation monitoring; partial mobile app data; and finally Action-Based data, which produce ODMs of the mobility demand according to city structures in terms of areas where people live and where they are interested in going to work, study, shop, etc., that can be extrapolated by using the POIs, the city structure and statistical information [23,24].

It should be remarked that if the Organization Data include static data, or more precisely information that slowly can change in time, other data (Sensors Data and Vehicle and People data) are for the most part information changing over time and thus they must be continuously acquired and elaborated to extract any typical behavior, and they must also be frequently updated since they represent a continuously evolving situation.

Due to the high variability of data formats and distribution modalities, there is an increasing interest in the exploitation of intelligent transportation systems (ITSs) and in the creation of National and Regional Access Points (NAP and RAP) and data spaces to be used to federate different data providers and to realize centralized access points to static and real-time data on mobility (see Section 3 for further details).

## 3. Mobility and Transport Data Formats and Standard

The mobility and transport data include a great number of different pieces of information represented with different standards and distributed with different modalities and protocols. In this section, a survey on the data formats and standards is provided.

### 3.1. Organization Data Group

Statistical data, including surveys and reports on population (census data), vehicle models and motorization, etc., can offer insights on the mobility and transport issue [25]. However, such kinds of data are sporadically updated and may be affected by missing or obsolete information [26]. Typically, statistical data are distributed as text files or Excel files, SDMX (Statistical Data and Metadata eXchange), CVS (Comma Separated Values) or JSON (JavaScript Object Notation) formats. Data retrieval can be performed manually or by using specific API to enable machine-to-machine access, for example, via FTP(s) or HTTP(s) download. In most cases, the statistical data refer to geographic areas at a high level such as regions, provinces, local administration, municipalities, etc., referring to other entities for their definition, i.e., the GIS.

GIS data are the main backbone of mobility and transport solutions since they can provide descriptions of urban, periurban and extra-urban infrastructures, modeling road networks, including POIs, buildings, road signs, semaphores, shapes of the buildings, administrative areas, etc. OpenStreetMap (OSM) [27] is nowadays one of the most developed services to create, update and distribute such data. Several web services are available to download the OSM data for the whole planet or for specific areas: for example, Geofabrik (https://download.geofabrik.de/, accessed 30 November 2023), HOT Export Tool (https://export.hotosm.org/, accessed 30 November 2023) and BBBike (https://download.bbbike.org/osm/ accessed 30 November 2023). The downloaded data come in OSM.PBF (Protocolbuffer Binary Format) files that can be processed with the Osmosis tool [28] or ingested into an on-premises installation of OSM. Alternatively, OSM data can be obtained with some limitations in Esri Shapefiles, GeoJSON, SVG, SQLite and other formats. Such data can be easily imported in GIS software like ArcGIS (version 10.6 and above) or the open-source QGIS (version 3.28.5 and above) for editing and inspection. Specifically designed to exchange static road attributes, e.g., speed limits, is the TN-ITS platform [29].

Services to render the GIS data described in the PBF files as tiled maps in PNG format are available; for example, the tile server released by Overv [30]. Raster images may include cadastral representations and satellite maps obtained from open services, such as Microsoft Bing, Google, Esri, OSM, Copernicus and Planet, just to name a few. Such data are usually available in PNG format, thus using the OpenGIS WMS/WFS (Web Map Service) interface standard over HTTP/HTTPS. Another kind of GIS data is the ground elevation information. Such data are exploited to compute the slopes of roads that could be taken into account particularly for soft mobility solutions; for example, to compute routing that avoids slopes greater than a given value. When this information is not included in the road graph, it can be extracted from Digital Elevation Models (DEMs)—or more precisely, Digital Terrain Models (DTMs)—created from aerial surveys by using LiDAR or other acquisition modalities, including satellite data. DEMs/DTMs are raster data that can be provided in different formats like Esri grid ASCII (ASC) files, which are textual representations of matrix data with georeferenced information defined in the header, or as GeoTIFF, an extension of the TIFF 6.0 file format to accommodate georeferences. GIS data are also provided by local government authorities, typically as Shapefiles or GeoJSON, describing information; for example, the positions of specific parking slots, streetlamps, urban furniture, reserved lanes, lane sizes, sidewalk sizes, etc. In this context, curb zones are gaining more and more attention as they are specific areas between the pedestrian realm and the road dedicated, for example, to the pick-up/drop-off of passengers and goods or as parking for sharing transports. The Curb Data Specification (CDS) [31] is a data standard and set of APIs under development realized by the Open Mobility Foundation for the management of curb zones. It includes static data, namely both positions and regulations of the areas, and real-time and historical data to communicate the events, occupancy, usage, etc. of curb zones. In a similar context, the Alliance for Parking Data Standards, an organization formed by the International Parking and Mobility Institute (IPMI), the British Parking Association (BPA) and the European Parking Association (EPA), is working on the definition of a standard and API for sharing parking data [32]. As for GIS data, the European INSPIRE directive [33] was developed in order to provide easy access to spatial data.

MOs’ data describe their transportation offers. For public transport services, such as buses, trams, trains, ferries, metro, etc., such data include static data—timetables, routes, lines, stops and prices—as well as dynamic data—related to real-time vehicle monitoring activities to provide, for example, any real-time positions of vehicles and their occupancy. The transportation offer information can be formalized and distributed by using different standards. The General Transit Feed Specification (GTFS) [34] is composed of two main parts: (a) the GTFS schedule represented as text files to describe static information (stops, routes, rides and prices) and (b) GTFS-RT (real-time) based on protocol buffers, a language-neutral, platform-neutral extensible mechanisms for serializing structured data in binary format, used to communicate trip updates, vehicle positions and service alerts. The Network Timetable Exchange (NeTEx) standard [35] is a general-purpose XML format composed of three main parts to describe the (a) public transport network, (b) scheduling and (c) fare information. It covers subsets of the European Public Transport Reference Data Model (Transmodel) [36], which is a conceptual model of common public transport concepts and data structures. NeTEx is comparable with the GTFS schedule standard, while the Standard Interface for Real-time Information (SIRI) [37], also based on the Transmodel principles, is analogous to the GTFS-RT to transmit real-time updates on the schedule data. It is worth noticing that in the scope of the European Transmodel, Open API for distributed journey planning (OJP) [38] has been devised to provide common interfaces to enable interoperability among multiple journey planner systems. Similarly, Telematics Applications for Passenger Services Technical Specifications for Interoperability (TAP TSI) [39] and RailML [40] data formats define technical and operational standards for the interoperability of the railway systems. When it comes to railway systems, the Open Sales and Distribution Model (OSDM) [41], based on the JSON format, is used to share information for ticket sales and distribution.

In the context of sharing transport-solution information, the most relevant standards, in addition to the NeTEx and SIRI that are designed to also consider the sharing mobility solution, are the General Bikeshare Feed Specification (GBFS) [42] and the Mobility Data Specification (MDS) [43]. The GBFS is composed of a series of JSON files reporting real-time data on stations, prices, vehicle availability, etc., while the MDS includes a set of APIs working on HTTP requests and delivering JSON responses. The MDS can be used to obtain real-time information as well as historical data: indeed, while the GBFS was designed to provide information for the FUs and is typically freely accessible, the MDS provides greater information addressing more specifically the PAs’ and MOs’ needs and usually requires specific authentication for data access. The GBFS and MDS seem to not yet be so widespread and adopted at the time of writing: indeed, even if several sharing MOs joined the standards, these data are not provided for all the cities where they operate.

Organization Data also encompasses information related to planned and unplanned events. Planned events can include temporary regulations/changes in the road structure issued by PAs that may alter the normal mobility environment. This includes car accidents, road network restructuring, a temporary traffic ban in specific areas, a change in road directions, the creation/closure of roads, etc. When available, these data are released by PAs or police offices as open data that can be exported as a CVS/XML/JSON text file or that can be retrieved by using dedicated API from some GIS. As for the other open data previously considered, the main problem is related to the variable availability with a scattered distribution over territories and limited interoperability since each provider releases data in different formats and with different APIs. These changes are not easily propagated into open services such as OSM and Google Maps.

In the context of urban mobility in smart cities, DIN SPEC 91367 [44] has been proposed as a guideline for data models, accesses and interfaces for any real-time traffic management promoting the exchange of information among mobility providers, city councils and city administrations and covering several domains like public transport, sharing mobility and infrastructure.

Logistics is another relevant domain in the smart mobility environment, in particular the last-mile delivery that has a relevant impact on urban traffic. In this context, we can find the Open Trip Model (OTM) [45], a data model used to share real-time information regarding trips for goods delivery. However, no other standard has seemingly been proposed so far. Indeed, working groups are still active in promoting digitalization and data sharing and interoperability in the logistic sector, such as, for example, the European Digital Transport and Logistics Forum (DTLF) [46] or the Australian National Freight Data Hub (NFDH) [47].

### 3.2. Sensor Data Group

IoT/IoE sensors are exploited to acquire real-time measurements and information on the mobility environment. Sensors can provide different kinds of data: traffic vehicle density, number of passages through specific areas, number of vehicles in parking areas, environmental information on weather, pressure, pollutants, bike flows, people counts and flows, etc. Most of the sensor devices can provide data in textual format; for example, using XML or JSON file formats. Data transmissions follow a client/server architecture, with push or pull modalities for sending/receiving messages. The most common pull protocols are REST calls over HTTP/HTTPS or FTP. Push protocols, which obtain data via event-driven subscriptions, include WebSocket (WS), Constrained Application Protocol (CoAP), Message Queue Telemetry Transport (MQTT), Advanced Message Queuing Protocol (AMQP) and FIWARE NGSI V2/LD [48,49]. TV cameras are a more particular case since the transmission of videos usually consists of a continuous data stream. Different codecs and video containers can be used (for example, the H.264 codec in an mp4 container) and several protocols are available, such as HTTP Live Streaming (HLS), Real-Time Messaging Protocol (RTMP), Web Real-Time Communications (WebRTC), Real-Time Streaming Protocol (RTSP) and Dynamic Adaptive Streaming over HTTP (MPEG-DASH). In the last years, affordable cameras endowed with a processing unit (even equipped with GPUs) capable of elaborating the video feed on the edge and extracting meaningful information have emerged. Information extracted on the edge by the TV CAM processing unit (e.g., the counting) can be transmitted as text messages such as JSON in MQTT/NGSI, thus reducing any privacy issue related to image transmission. Cameras are mainly used to count persons [15,50] or vehicles (in some cases called virtual spires) staying in specific areas or passing through particular gates. Tracking applications to assess mobility patterns is possible by using, for example, person reidentification or car plate reading [51]; this approach is also used to estimate the average velocity in highway segments and produce tickets. However, the usage/distribution of this kind of information can be limited by privacy regulations, and for this reason, other means to estimate origin–destination trajectories should be preferred. An alternative can be processing car license plates to compute some unique hash codes for the day/hour, which can be used to match the origin and destination of a vehicle performing some trip within the same day/hour.

Note that some of this traffic-flow-data-related information could also be retrieved from companies providing data via API and/or web services. For example, HERE or Waze provides APIs to obtain traffic feeds and also includes information on events like road incidents. However, these services typically require paid subscriptions. Most of those data services obtain data by merging several different data sources, which include sensors, on-board units (OBUs), mobile apps and in some cases also cellular data provided by a mobile telecom operator, helping to represent the demand of mobility.

Open data on traffic flows can be available for some areas; GraphHopper offers a list of open data on traffic information [52]; however, the coverage is sparse/scattered and does not always provide real-time information. Such data, in particular from European providers, are mostly transmitted by using the DATEX-II standard [53] based on the XML format, which has grown from a standard for the exchange of traffic-related data to a coherent set of standards supporting the digitalization of the entire road-transport ecosystem including information such as the roads, areas, itineraries, abnormal traffic, accidents, obstructions, variable message signs, measured and elaborated data, parking, traffic management and traffic signal management.

It is noteworthy that smart mobility systems are not limited to sensors to perceive and collect information on the environment and can also include actuators or hybrid (sensor and actuator) devices to perform actions in the environment. For example, smart traffic lights and variable message signs (VMSs) (dynamic signage) are used to implement Adaptive Traffic Signals Control (ATSC), which is a dynamic traffic-management strategy actuated by controlling traffic signs according to the actual traffic demand. The DATEX-II standard and the National Transportation Communications for Intelligent Transportation System Protocol (NTCIP) standards [54]—a family of standards designed to enable interoperability and interchangeability among computers and electronic traffic control equipment—can be used to operate on actuator devices. In other cases, the traffic flow density is adopted to control road luminaries in the context of city smart light. This approach should be conformant with the UNI11248:2016 standard [55].

### 3.3. Vehicle and People Data Group

The Vehicle and People data group includes data that are mainly produced by FUs’ devices; therefore, they are the primary source to estimate the demand for mobility that both PAs and MOs have to satisfy.

Given the diffusion of mobile devices such as smartphones and tablets, they can be used to estimate the mobility patterns of users [56]. Cell tower connections, provided by telecommunication operators, give a rough representation of the FUs’ movements. Since mobile cell towers are deployed to improve the device connection quality and not to track any users’ movements, they are placed with a nonuniform mesh and can cover wide areas with some needed overlap among different towers to guarantee some service redundance. In addition, the data resolution that can be sold by a telecom operator is typically related to the size and shape of the census areas, which are unregular on the territory, as it occurs with the distance among towers. Indeed, a smartphone during a certain period could connect to different cell towers, even if its user does not move in different places, and thus the connection can jump among towers in a very short amount of time, giving the impression of a fast jump by the user among different points inside the territory. On the other hand, users could travel in a neighborhood and apparently remain connected to the same tower position. For these reasons, locations coming from connection data are useful to estimate travel patterns in extra-urban scenarios (e.g., moving from one city to another) and are less helpful to understand urban movements at the micro/meso scale. A micro-scale representation of the FUs’ mobility could be provided by the geolocalization functionality of the mobile apps installed on the user’s devices that typically exploit GPS information. With signed consent by the user to be tracked according to the GDPR [22], an app can record and transmit any device position, providing individual trajectories to be used for the collective estimation of origin–destination matrices. However, to be effective, the tracking app must be exploited by a relevant number of users (from different categories), thus requiring adequate infrastructure to guarantee a functional service and to handle huge data and a compelling service useful to users. For example, routing and travel planning apps like Google Maps (by Alphabet Inc., Mountain View, CA, USA), Waze (by Waze Mobile Ltd, Palo Alto, CA, USA), Apple Maps (by Apple Inc., Cupertino, CA, USA), Moovit (by Moovit Inc., Ness Ziona, Israel), HERE WeGo (by HERE Global B.V., Eindhoven, The Netherlands), etc., and also Facebook (by Meta Platforms Inc., Menlo Park, CA, USA), can be/are equipped to acquire such a kind of data that are later on made available. When requiring users to review their tracked movements and to provide additional information (for example, describing their motivations to travel), thus realizing the so-called travel diaries, more detailed data are acquired from paid users using smartphone apps by specialized companies. Both data from telecommunication operators and app vendors are not released for free and have to be purchased, and even if acquired in real time (this explains why they are depicted in the real-time group of Figure 1), they are usually distributed as collective historical data in an aggregated way to protect user privacy through dedicated API used to perform queries on the vendor datastore. No particular standards are used, and data are retrieved in formats defined by the vendors. These motivations limit usability in the realization of smart mobility platforms.

An interesting alternative is the exploitation of the data recorded from MaaS applications. MaaS systems centralize all the mobility offers and become the main access point for the FUs that search for travel solutions. Therefore, for their purpose, they need to acquire and manage detailed information on the offer and receive from the final users their multimodal travel demand (with some limitations on the kind of user profile, depending on the activated MaaS offer). In the case of a MaaS Open-Platform business model, where the platform is managed by public organizations, travel data could be shared with public and private MOs or released as open data (with adequate protocols to guarantee user privacy), even in real time. Currently, most MaaS solutions are in development or experimental phases and are not yet widely deployed/widespread, while some relevant effort has been carried out on this topic by national and international organizations [57,58,59]. Moreover, the Transport Operator Mobility-as-a-service Provider (TOMP) API [60] has been proposed as a standardized way of communication in the context of MaaS to handle operations from trip planning, to booking, to trip execution and payment.

A similar result can be obtained by exploiting OBUs installed on vehicles, for example, for insurance purposes, to reduce risks and the uncertainty of possible event occurrences by fleet operators to control the vehicle status and positions and in some cases also to collect information on both driving styles and conditions. More innovative systems on board also include OBUs involved in the context of autonomous vehicle management and communications. Some of those systems enable vehicle-to-everything (V2X) connectivity, i.e., including vehicle-to-vehicle (V2V), vehicle-to-infrastructure (V2I) (also for communicating with semaphores to reduce the time to travel and tuning velocity to reduce possible stop and go at semaphores, thus also reducing emissions), vehicle-to-pedestrians (V2P) and vehicle-to-network (V2N) communications. Such technologies are fundamental in the development of CCAM and CAV solutions, where vehicles share local information with both ITS and all-road users, using, for example, messages in the ETSI ITS standard [61]. While the aim of such solutions is to perform coordinated or automated actions, for example, the anticipated cooperative collision avoidance (ACCA) service [62,63], the data produced and shared by vehicles could also be used to perform analyses and predictions on the mobility environment since they include information on vehicle status (e.g., velocity and position), measurements of the local environment, signals of unexpected events, etc. Dedicated protocol stacks like the OpenC2X [64] and Vanetza [65] have been devised to exchange data in XML, JSON or Abstract Syntax Notation One (ASN.1) [66] formats. Other standards related to the communication of in-vehicle sensors are SENSORIS [67], the Extended Vehicle (ExVe) [68] and the Open Diagnostic Data Exchange (ODX) [69]—also known as MCD-2 D and designed for the exchange of diagnostic data.

Smartcards, used to pay for travels on public transport services, can provide relevant information [70], in particular if entry–exit Automated Fare Collection (AFC) systems are used, where any FUs have to pass the card when getting on and off the vehicles. For example, Oyster smartcards in London track the metro entrance station, some passages and the exit by forcing travelers to insert the ticket or the card to go through each in/out gate. In this way, both the origin and destination of each traveler can be recorded. These data are acquired by MOs, mostly working in public transportation, and released as open data by using text formats like JSON, XML, CVS or the Keyhole Markup Language (KML).

Social media platforms produce huge amounts of data, whereby a number of the posts produced by users may contain location information and can be used for smart mobility tasks [71]. For example, Telegram is gaining importance as a channel to quickly communicate news and events from several organizations to FUs who can interact and respond, while X (formerly Twitter) has closed their open API, losing a lot of attraction and possibilities on the exploitation of social media data for mobility and many other aspects [72]. Moreover, when legally possible, web-scraping tools are available to extract data from social media and deliver them in JSON, CVS, HTML and Excel.

### 3.4. Data Spaces and National Access Points

Although the previous sections described mobility data, it should be remarked that most data are provided in different formats/models by different providers and distributed with different modalities. Therefore, building a system able to collect such data is a very complex task [73]. On the one hand, all different standards and distribution modalities must be considered, thus requiring the development of specific acquisition procedures with a high flexibility. On the other hand, the activity of finding correct data providers and studying their distribution modalities is time consuming and prone to risks of missing some needed information for the purpose planned. In the last years, national and international public organizations have been trying to design and implement data hubs or dataspaces [74,75] capable of integrating the different data providers by defining interoperability standards to realize unique access points for the distribution of mobility and transport data, as well as data spaces in other domains, which are out of the scope of this paper. In Europe, the first project aimed at the development of a mobility dataspace started in 2019 in a German consortium [76,77,78] proposing a decentralized and distributed system with a directory to publish available data sources and services (i.e., a data marketplace) and a vocabulary provider that gives information on standards and APIs (e.g., DATEX-II, NeTEx and SIRI). To extend such a solution to the whole European area, the European Commission started in 2022 with two main actions. The concept of a datahub is not far from the early implementation of the API collection, which was proposed by E015 for Regione Lombardia for the Expo 2015 [79].

The National Access Point Coordination Organization for Europe (NAPCORE) project [80] was kicked off to promote and coordinate the creation of NAPs in all the EU Member States to fulfill the European Intelligent Transport System directive (2010/40/EU) [81]. Conversely, the PrepDSpace4Mobility [82] project was launched to map data ecosystems and propose common building blocks to create a single European market for mobility data. On a global scale, the International Data Spaces Association (IDSA) [83] is working for more awareness and the global adoption of data spaces to ensure the wide access and usage of data. In the logistics and freight domain, the FEDeRATED project [84] aims at laying the foundations for the development of an interoperable data-sharing infrastructure for B2A (business to administration) and B2B (business to business) data exchange.

As for the necessity of increasing interoperability in the more general context of smart cities, it is worth reporting the Minimal Interoperability Mechanisms (MIMs) [85], which are able to provide a set of tools to develop interoperability mechanisms for data, services and systems.

Moreover, legal aspects concerning data acquisition, exploitation and distribution must be considered, since to perform each of those activities, a specific grant has to be legally acquired/received [86]. Data sovereignty and trust indeed play a fundamental role in creating common data spaces to share the data produced by different organizations or individuals. Such regulations are fundamental to guarantee that the data-producing entity could maintain its authority and be able to control the exploitation of its data, therefore encouraging the distribution of proprietary information.

On the other hand, such regulations provide the consumer with a clear understanding of the data-usage rights (and costs) enabling the definition of an effective business model without the risk of legal issues. It is important to note that such regulations can vary from country to country. For example, since 2018, the European Union has adopted the General Data Protection Regulation (GDPR) [22], and in 2022, the Data Act [87] was adopted, complementing the Data Governance Regulation of 2020. Differently, in the United States, several laws on data protection were enacted both at federal and state levels. Therefore, there are several data-protection regulations that may vary for domains and countries. For example, in the context of mobility, the Driver’s Privacy Protection Act of 1994 describes the privacy and disclosure of personal information gathered by state Departments of Motor Vehicles. More generally, at a federal level, the Federal Trade Commission Act of 1914 empowered the U.S. Federal Trade Commission [88] to enforce privacy regulations, while at the state level, we can name the California Consumer Privacy Act of 2018 or the Virginia Consumer Data Protection Act of 2021. Consequently, any data space proposing itself as a marketplace for mobility data is therefore forced to design and implement solutions to handle data property, defining trust mechanisms and providing methods to delegate data usage to third parties. In this context, the iShare foundation [89] offers different types of agreement (functional, technical, operational and legal) that participant organizations following the iShare schema must agree with by integrating API based on well-grounded authentication technologies, such as the OAuth [90] or OpenID Connect [91]. In this way, any participant can decide which data are made available to which partner, when and for which purpose, therefore maintaining full control over the data. Together with the FIWARE foundation [48] and FundingBox [92], iShare has launched the i4Trust program [93] aimed at facilitating the creation of data spaces: FIWARE contributes to interoperability aspects and iShare covers both trust and legal features, while FundingBox is devoted to funding acquisition, so as to help the implementation of the FIWARE and iShare building blocks.

To summarize, in Figure 2, we reported a Venn diagram highlighting the coverage of available standards with respect to the main mobility and transport aspects: vehicles, public transport, sharing mobility and logistics. As can be seen, some of them are specific for a given aspect while others cover multiple ones. Most standards aim at data exchange among different operators and PA (infrastructure), and a relevant number of standards has been proposed and adopted by Public Transport and Sharing solutions. More recently, attention has been focused on data modeling and exchange among vehicles and infrastructure.

To have a more detailed view of the available data and standards for mobility, in Table 1, we reported such mentioned standards, providing, for each of them, information about the temporal domain (static, historical and real time), the main mobility domains (the same as those in Figure 2) and some more specific subdomains. Additionally, we included information on the data formats used by these standards. In the last row, the number of entries covered in the column is reported. Such information can give some insights into the most addressed aspects by current mobility standards. At first, we can observe that most standards are devoted to static or real-time data, while historical data are less considered. In our opinion, this is reasonable, since accumulating historical data and possibly performing an analysis on them is a responsibility belonging more to mobility platforms than to data standards, which in most cases are related to the real-time consumption of data for creating services for final users. Regarding mobility domains, less supported standards are the ones on logistics and vehicles. Indeed, logistics has recently gained increasing attention in the mobility context, while single-vehicle standards are still fragmented and not widely diffused due to the still-limited adoption of OBU, CAV and CCAM technologies. Road network information (both static, as any road graph description, and real time, as traffic, car accidents or congestion) and information about public transport are the most developed subdomains, probably due to the ever-increasing investments in smart cities that mostly stake on public transports and infrastructure management to create greener and sustainable mobility solutions. Finally, most standards employ XML or JSON formats since they are at the same time human and machine readable and can allow for the development of API with limited effort.

## 4. Data Management and Exploitation

The efficient management and utilization of diverse data types/models have become imperative in the domain of mobility and transport. Various data-modeling methodologies play a pivotal role in organizing and understanding the intricate relations among vast and complex datasets associated with urban transportation systems. These methodologies range from traditional relational data models to innovative reticular-, graph- and ontology-based approaches, which offer distinct strategies for structuring, storing and analyzing complex data graphs. Each approach brings unique strengths and challenges, catering to different aspects of the dynamic transportation landscape, including real-time traffic information and thus time series, spatial data, environmental variables and social media insights. Understanding the intricacies of these data-modeling methodologies is crucial for developing comprehensive data-management strategies that can effectively support intelligent transport systems and optimize traffic flow, thus permitting fast constrained routing and facilitating sustainable urban planning and development.

Relational data modeling involves organizing data into tables with rows and columns, where each row represents an entity while columns represent attributes of that entity; they can be also called SQL databases [94]. Relationships among entities are established through keys and foreign key constraints [95]. The major benefit of this type of data-modeling technique is that it ensures data consistency and accuracy, critical for maintaining reliable mobility data. This allows transactions to be processed reliably and consistently, which is vital for handling critical real-time transportation data and ensuring ACID Compliance [96]. The relational database is a well-established technology with widely used methods to manage various forms of transportation data. Vaisman et al., in [97], based their trajectory mobility data warehouse on relational databases. This solution allows one to handle temporal queries integrating relational spatial and nonspatial data. Relational database modeling can effectively manage spatial data. An example is reported in [98], where Etienne et al. described how to set up spatiotemporal relational datasets to model Maritime traffic exploiting via PostGis and Qgis and also spatial data representations and querying. This methodology could be challenging when dealing with large volumes of real-time data as relational databases may have limitations in handling a rapid data influx for the indexing time. On the other hand, relational modeling technology is not sufficiently flexible for handling complex and evolving data relationships, a characteristic that could be a requirement when dealing with diverse mobility data sources allocated on complex graphs. Complex queries and large datasets [99] may lead to performance issues without appropriate indexing and optimization, hindering real-time data-processing capabilities.

Document-oriented data-modeling approaches are built on the BASE (Basically Available, Soft State and Eventually Consistent) model [100], which emphasizes scalability and provides a flexible schema [101]; they are noSQL databases. Kanojia et al., in their work [102], proposed guidelines for the migration from a mobility solution in the context of smart cities based on relational databases toward a NoSQL when dealing with large volumes and big data [103]. Document-oriented data modeling involves the use of documents (typically in JSON or XML format) to represent data. Each document contains data attributes and values, and collections of documents are organized within a database [104]. Nurhadi et al., in [105], proposed NoSQL databases for building a smart city data lake stressing the capability of handling diverse data types, such as weather data, environmental data and social media data, thanks to its capacity to manage semistructured and unstructured data effectively without indexing at their early data ingestion. Authors have also evaluated performances, scalability, accuracy and complexity in terms of constructing specific queries. MongoDB and Redis achieved the overall best scores, and the authors concluded that NoSQL databases compromise consistency to provide high performance and scalability. Cheng et al. [106] instead chose, for smart city big data platforms, CouchDB as the primary storage solution. Researchers have focused their solution on NoSQL data and highlighted the flexibility in terms of scalability and in handling the evolution of data structures over time, thus enabling the seamless integration of new data formats and attributes. This aspect is very important when dealing with mobility data since transportation data and relationships may evolve over time; for example, the evolution of GTFS data in the different seasons. The authors have affirmed that some issues need to be considered for a more advanced smart city big data platform; for example, how to enhance the support of semantic data and how to share knowledge among different applications in terms of consistency. Document-oriented databases might pose challenges in maintaining data consistency, particularly for critical mobility data processing [107], since they are not supported by sophisticated indexes to maintain complex relationships.

Ontology-based data modeling involves the creation of a formal representation of knowledge within a specific domain. It defines the concepts within the domain and the relationships among concepts, providing a semantic framework for understanding and organizing data relationships [108]. It is a conceptual model of a part of reality, made of concepts linked in a given application domain [109]. Semantic Representations enable a precise and formal representation of the relationships among various data entities, creating a graph of entities, which may actually also have instances and where you can pose semantic queries defining the searched semantic relationships. They are formalized in SPARQL [110]. Komninos et al., in [111], highlighted its effectiveness when modeling ontologies for smart cities, especially in the field of transport and mobility, which constitute a high-priority domain toward a deeper understanding of complex transportation data. Ontologies are grounded on a set of vocabularies (e.g., SAREF, OTN, WGS84, FOAF, etc.) and facilitate data integration from diverse sources by establishing a common understanding of the domain/subdomain, promoting seamless data sharing and interoperability among different transportation systems [112]. For Jin et al., any mobility data integration requires the development of conceptual frameworks that support appropriate data-representation and -manipulation capabilities [113]. They proposed a conceptual model for trajectories in urban spaces via semantic data representation. Ontology-based data modeling is instrumental in representing and managing diverse mobility data types, including GIS data, real-time traffic from IoT, sensors, environmental data, GTFS, parking, etc. [114]. It enables a comprehensive understanding and modeling of complex transportation relationships, thus facilitating effective traffic management, route optimization and the development of intelligent transportation systems for sustainable urban mobility. Codescu et al., as documented in [115], structured the semantic geometadata associated with OpenStreetMap by means of an ontology known as OSMonto. The OSMonto ontology can serve as a reference model to enhance the semantic interoperability in relation to the geodata modeling, a fundamental aspect in the context of mobility and transport. Katsumi et al. proposed an ontology specific to the transportation concepts related to parking and travel activities named ICity [116]. Bellini et al., in [117], proposed the ontology named Km4City (Knowledge Model for the City) to manage big data coming from a variety of sources in the context of smart cities, including the modeling of mobility and transport, graphs, public transportation offers, services available on the roads, traffic sensors, parking, sharing, cycling, attraction points, services for city users, etc. On the other hand, Bellini et al. [117] also highlighted that ontologies are unsuitable/inefficient to effectively handle time series data, a crucial aspect in the context of mobility and urban infrastructure, and thus proposed Elastic Search and then OpenSearch to cope with them.

It becomes evident that there is no one-size-fits-all storage model. The inadequacy of ontologies for handling time series data underscores the necessity for alternative approaches in data management. Traditional data lakes, while serving as repositories for diverse datasets, fall short in terms of data integration to serve the front end and data distribution in real time driven by queries on the basis of complex relationships. When immediate integration is required for service delivery by using different data sources, the cost and complexity of reintegrating data from data lakes on the fly at the moment of data consumption can be prohibitively high. This highlights the inherent limitations of data lakes. On the contrary, data warehouse solutions allow for structured storage and the efficient retrieval of data while providing a performant foundation for data consumption across various contexts. This approach has turned out to be particularly effective in addressing any challenges posed by time series data, enabling the seamless integration of different data sources and supporting complex queries with spatial, relational and temporal dimensions. Badii et al., in [118], extended their contributions by adopting a data-warehousing approach, leveraging the foundational ontology (Km4City) in proposing a Snap4City solution that integrates ontology and Open Search for time series management. This approach allows for structured storage and efficient data retrieval, providing a performant foundation for data consumption across various contexts, such as data analytics, simulation and artificial intelligence. The solution facilitated the seamless integration of diverse datasets, empowering the system to execute complex queries with spatial, relational and temporal dimensions, which is crucial in the context of mobility and transportation applications; for example, to search for specific services along public transportation paths, to support constrained routing, etc.

### Data Flow Diagram for Mobility and Transport Analysis and Services

Mobility and transport data can be exploited to (i) produce other kinds of data models, as well as to (ii) enable the implementation of a large range of applications.

More precisely, Figure 3 represents a data flow diagram showing, from the left represented as green blocks, that original elementary data sources in different standards and formats (such as GIS, POI, traffic flow sensors, social media, etc.) can be used to produce more complex data models/sets to understand mobility and transport conditions (such as orthomaps, parking status, mobility demand, origin–destination matrices, transportation offers, etc.). All these data, both elementary and elaborated/derived, can be used for the operative management of mobility and transport in the city, as well as for the simulation and optimization (to perform a tactic and strategic plan). Such processes primarily involve data-transformation tools to extract from basic data sources a set of more usable and valuable data models, which are actually more usable for deeper analysis. The operative management processes in the blue blocks include, for example, estimations of the data (listed in pink blocks) such as:KPIs (key performance indicators) such as Sustainable Urban Mobility Indicators (SUMI) [119] and the SUMP, Sustainable Urban Mobility Plan [120], required to assess city mobility and transport management conditions/facilities;Predictions of traffic flow [121], parking lots status [16], sharing service conditions, etc., which are typically produced by some deep learning models;Anomaly detections: for example, comparing real-time conditions with respect to typical or predicted conditions and thus producing notifications, tickets for maintenance and alarms when critical conditions/events are detected;Routing, multimodal routing and conditional routing for producing routing paths by taking into account real-time traffic/environmental conditions or possible changes inside city structures due to last-minute ordinance, accidents and natural/non-natural events;Origin–destination matrices (from census data, from OBU devices, from mobile apps data, from mobile operators’ data, etc., or by data fusion): trajectories for people and vehicles, semaphores cycles and simulations, in general;Prescriptions to solve critical conditions, such as improved semaphore cycles to reduce time to across the city, changes within city viability, etc. They are typically produced by using operative research algorithms exploiting optimization models.

Such data results and evaluations can be used to perform and assess different mobility and transport scenarios. Different scenarios may imply and produce corresponding traffic conditions, pollutant emissions, parking status, public transportation loads, etc. Traffic scenarios can be identified on the basis of different mobility data in some hypothetical, typical or future contexts by using corresponding data, see Snap4City scenarios [9]. Such data results are at the basis of the what-if analysis approaches (orange blocks) to assess and set up different strategies for sustainable mobility, which can solve specific critical or hypothetical conditions in order to improve mobility and transport scenarios. Finally, all the data, results and information are presented to operators (and maybe to final users) by means of operative dashboards [122] as well as aggregated and integrated into smart city Digital Twins representations [123,124] exploiting client-side business logic [125] to guarantee seamless interactivity.

## 5. Conclusions

The growth of data sources and models in the mobility and transport domain has strongly proliferated in recent years. Most of them provide a large range of overlap, and the most recent formats were created just to add new information to enable new solutions and business models. We hope that this work of analysis could be useful for many researchers that have to identify needed data among the several formats and relationships among them. We analyzed, in this paper, a huge amount of data formats and models, classified as Organization Data, Sensors Data and Vehicle and People data. The classification is a conceptual model, which allowed us to describe data model relationships with respect to higher-level information, which can be exploited on the basis of innovative applications. To this end, a data flow was developed to describe the main principle behind the process allowing it to perform Operational Management, as well as simulation and optimization processes, which are on the basis of the what-if analysis for controlling and planning changes in the mobility and transport infrastructure. This work was developed in the context of the CN MOST, a national center of sustainable mobility in Italy, by exploiting the Snap4City platform.

## Figures and Tables

**Figure 1 sensors-24-00441-f001:**
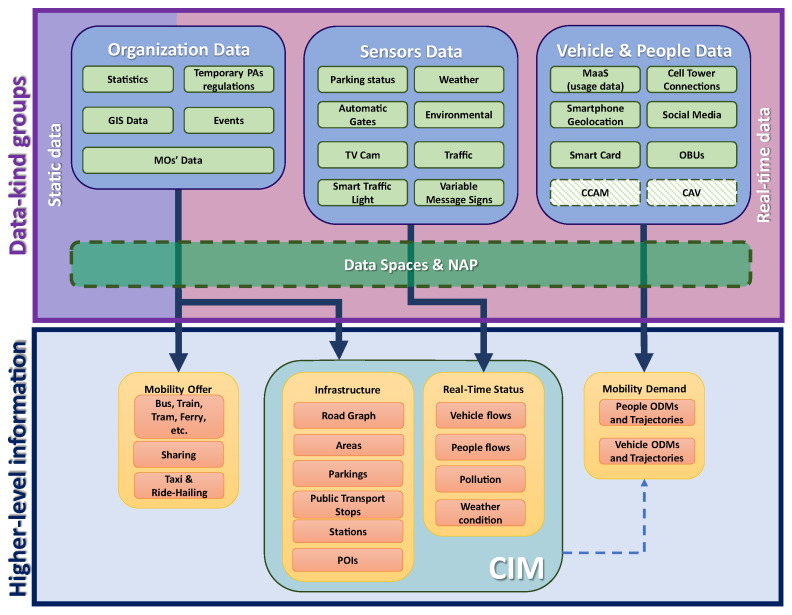
Schematic representation of the most relevant data-kind groups and derived higher-level information for smart mobility.

**Figure 2 sensors-24-00441-f002:**
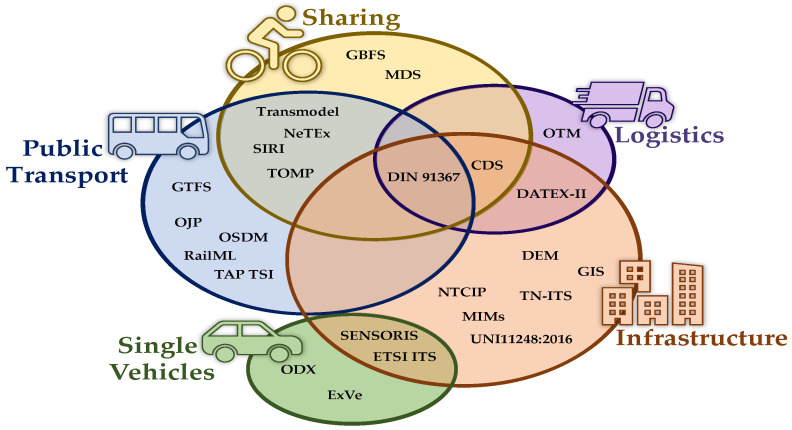
Venn diagram of standards for smart mobility domains. References: GIS data (OSM) [27], TN-ITS [29], CDS [31], GTFS [34], GTFS-RT [34], NeTEx [35], SIRI [37], Transmodel [36], OJP [38], TAP TSI [39], RailML [40], OSDM [41], GBFS [42], MDS [43], DIN SPEC 91367 [44], OTM [45], DATEX-II [53], NTCIP [54], UNI11248:2016 [55], TOMP [60], ETSI ITS [61], SENSORIS [67], ExVe [68], ODX [69].

**Figure 3 sensors-24-00441-f003:**
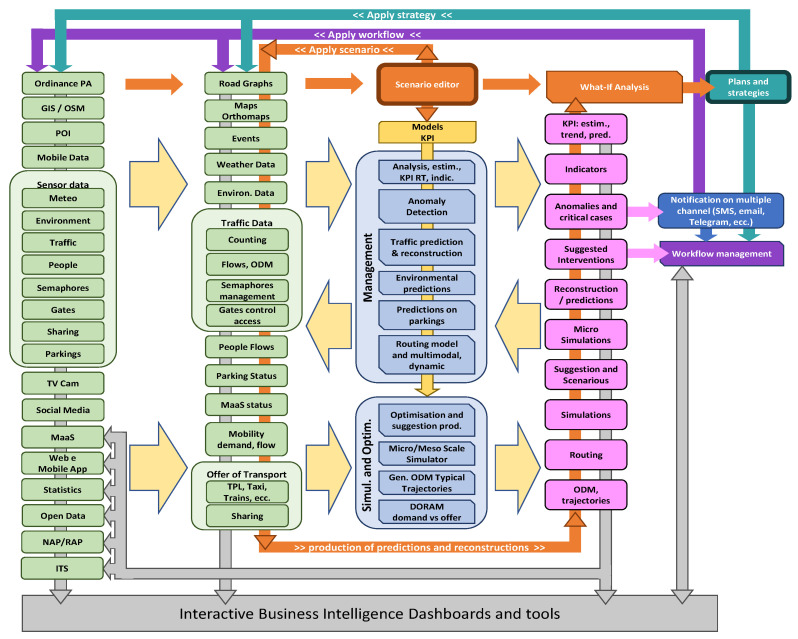
Conceptual flow diagram for ITS services according to a suitable mobility data model.

**Table 1 sensors-24-00441-t001:** Summary of data and standards for mobility.

Data and Standards	Temporal Domain			Mobility Domain					Mobility Subdomain														Format														
	Static	Historic	Real-Time	Infrastructure	Logistic	Sharing	Public Transport (PT)	Single Vehicles	Census	Road Network	Urban Elements	Traffic Signals	POI	Buildings	Terrain	Weather	Pollution	PT Urban: Bus, Tram, …	PT: Railways	Journey Planning	User notification	Vehicle Status/Diagnosis	Excel	SDMX	XML	CSV	JSON	GeoJSON	Protocol Buffers (PBF)	Esri Shapefiles	SVG	SQLite	RDF	PNG	GeoTIFF	Esri Grid ASCII (ASC)	ASN.1
Statistical data	X	X		X	X	X	X	X	X														X	X		X	X										
GIS data (government)	X			X						X	X		X	X														X		X							
GIS data (OSM) [27]	X			X						X	X	X		X	X													X	X	X	X	X		X			
TN-ITS [29]	X		X							X															X												
DEM (DTM, DSM)	X													X	X																			X	X	X	
CDS [31]	X	X	X	X	X	X				X	X																X	X									
GTFS [34]	X						X											X											X								
GTFS-RT [34]			X				X											X											X								
NeTEx [35]	X					X	X											X	X						X												
SIRI [37]			X			X	X											X	X						X												
Transmodel [36]	X		X			X	X											X	X	X																	
OJP [38]			X				X											X	X	X					X												
TAP TSI [39]	X		X				X												X	X					X												
RailML [40]	X		X				X												X	X					X												
OSDM [41]	X						X												X	X							X										
GBFS [42]	X		X			X																					X										
MDS [43]	X	X	X			X																					X										
DIN SPEC 91367 [44]			X	X	X	X	X			X	X		X					X	X	X					X	X							X				
OTM [45]			X		X																				X		X										
IoT/IoE Sensors—TV Cam			X	X						X			X			X	X								X		X										
DATEX-II [53]			X	X	X					X	X	X	X												X												
NTCIP [54]			X	X								X													X												
UNI11248:2016 [55]			X	X							X																										
TOMP [60]	X		X			X	X											X	X	X							X										
ETSI ITS [61]			X	X				X		X											X	X															X
SENSORIS [67]		X	X	X				X		X											X								X								
ExVe [68]			X					X														X			X		X										
ODX [69]			X					X														X			X												
	15	4	21	11	5	9	12	5	1	9	6	3	4	3	2	1	1	8	9	7	2	3	1	1	13	2	9	3	4	2	1	1	1	2	1	1	1

## Data Availability

Not applicable.

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
