# Peer review of "Data Sources and Models for Integrated Mobility and Transport Solutions"

_sensors, 2024, doi:10.3390/s24020441_

Round 1

Reviewer 1 Report

Comments and Suggestions for Authors

The main elements are: an overview of data sources and models, a review of standards and formats, a discussion of data models, a description of data management and utilization, and conclusions. Overall, this paper provides an extensive discussion of the large number of data formats and models in the mobility and transportation domain, as well as the associated data modeling solutions needed to implement intelligent mobility and transportation frameworks, analytics, and decision support systems that can help policy makers and mobility players face current and future challenges. The paper also has the following points that need minor revisions:

1. Figure 1 appears on page 2, but the actual description of what it contains is on page 4, so it is suggested that the position of the figure be adjusted to make the article more coherent.

2. Figure 2 Wayne's diagram is on page 9, but the explanation is on page 12, which should be adjusted.

3. Figure 3 has the same problem.

4. The third contribution mentioned in this paper is the observation of future trends, challenges and directions for the future, which should be described in more detail what kind of challenges and directions are proposed and appropriate recommendations should be made.

Author Response

[reviewer 1]

The main elements are: an overview of data sources and models, a review of standards and formats, a discussion of data models, a description of data management and utilization, and conclusions. Overall, this paper provides an extensive discussion of the large number of data formats and models in the mobility and transportation domain, as well as the associated data modeling solutions needed to implement intelligent mobility and transportation frameworks, analytics, and decision support systems that can help policy makers and mobility players face current and future challenges. The paper also has the following points that need minor revisions:

ANSWER: thanks!

  1. Figure 1 appears on page 2, but the actual description of what it contains is on page 4, so it is suggested that the position of the figure be adjusted to make the article more coherent.

ANSWER: the position of the Figures has been corrected.

  1. Figure 2 Wayne's diagram is on page 9, but the explanation is on page 12, which should be adjusted.

ANSWER: the position of the Figure has been corrected. The figure and its description improved.

  1. Figure 3 has the same problem.

ANSWER: the position of the Figure has been corrected.

  1. The third contribution mentioned in this paper is the observation of future trends, challenges and directions for the future, which should be described in more detail what kind of challenges and directions are proposed and appropriate recommendations should be made.

ANSWER: Thanks for the comment which allows us to better focus the paper. As reported in the new version of the paper, the main contributions are to provide:

  • an updated and comprehensive overview (although not exhaustive) of research and industrial literature about data modeling and types for smart mobility and transport.
  • main relationships among different data model concepts to highlight what kind of information (data models) can be obtained by processing them, in the whole value chain of data on mobility and transport scenario.
  • insights which can be derived by the data models and the business and derived in context of integrated smart mobility and transport systems addressing multiple data model, spaces and types.

Reviewer 2 Report

Comments and Suggestions for Authors

1. This manuscript reviews data sources and models for integrated mobility and transport solutions; however, it is disorganized and cannot provide an effective schema for related knowledge.

2. The related description cannot correspond to the figures of the schematic representation.

3. Giving some tables to show the related techniques and standards is better.

4.  In Section 4, data management refers to different databases, and the presentation of data modeling is confusing.

Comments on the Quality of English Language

Written English should be polished.

Author Response

  1. This manuscript reviews data sources and models for integrated mobility and transport solutions; however, it is disorganized and cannot provide an effective schema for related knowledge.

ANSWER: A clarification has been provided in the paper organization.

  1. The related description cannot correspond to the figures of the schematic representation.

ANSWER: The position and descriptions of the figures have been improved. A new table has been added to better clarify the comparison and  assessment model adopted.

  1. Giving some tables to show the related techniques and standards is better.

ANSWER: Thanks for the comment. A new Table has been introduced, Table 1. It puts in evidence and compares the several standards considered with respect to the temporal domain, mobility domain/aspects and subdomain/aspects, and formats. A corresponding paragraph has been added to explain the table. This also allowed to improve the understanding of the Venn diagram reported in Figure 2.

  1. In Section 4, data management refers to different databases, and the presentation of data modeling is confusing.

ANSWER: The section has been strongly improved, by adding more details and examples.

Reviewer 3 Report

Comments and Suggestions for Authors

This review article provides a collated discussion of data related to the mobility domain, as well as the associated data modeling solutions required to use such information to implement smart mobility frameworks, analytics, and decision support systems.

The article is well written, a wide range of sources is presented, on the basis of which a classification of methods and models is carried out.

Here, the article reviews the research literature on modeling and data for smart mobility and transportation. It is important that the main relationships between different data model concepts and their processing in the mobility and transport data value chain are indicated.

For domain researchers, the article is useful for understanding recent trends, open technical challenges associated with implemented data modeling techniques, as well as future directions that need to be addressed when implementing assistive technologies in the context of integrated intelligent mobility and transport systems, covering a set of models, spaces and data types.

The article may be accepted as is.

Author Response

[reviewer 3]

This review article provides a collated discussion of data related to the mobility domain, as well as the associated data modeling solutions required to use such information to implement smart mobility frameworks, analytics, and decision support systems.

 ANSWER: thanks

The article is well written, a wide range of sources is presented, on the basis of which a classification of methods and models is carried out.

  ANSWER: thanks

Here, the article reviews the research literature on modeling and data for smart mobility and transportation. It is important that the main relationships between different data model concepts and their processing in the mobility and transport data value chain are indicated.

 ANSWER: thanks

For domain researchers, the article is useful for understanding recent trends, open technical challenges associated with implemented data modeling techniques, as well as future directions that need to be addressed when implementing assistive technologies in the context of integrated intelligent mobility and transport systems, covering a set of models, spaces and data types.

 ANSWER: thanks

The article may be accepted as is.

 ANSWER: thanks

The language has been revised.

Round 2

Reviewer 2 Report

Comments and Suggestions for Authors

The quality of the figures should be improved.

Comments on the Quality of English Language

The English language should be checked carefully.